# The Impact of Nutritional Therapy on Gastrointestinal Motility in Older Adults

**DOI:** 10.3390/healthcare11212829

**Published:** 2023-10-26

**Authors:** Yohei Okawa

**Affiliations:** Department of Psychosomatic Medicine, Tohoku University Graduate School of Medicine, 2-1 Seiryo-Machi, Aoba-Ku, Sendai 980-8575, Miyagi, Japan; yohei-tky@umin.ac.jp; Tel.: +81-805483393

**Keywords:** elderly, nutritional therapy, gastrointestinal motility, functional digestive disease FGID, daily lifestyle habits

## Abstract

Japan is becoming a superaged society, and nutrition therapy for the elderly population is very important. Elderly individuals often have multiple diseases and are prone to malnutrition. Furthermore, functional constipation, diarrhoea, faecal incontinence, etc., may occur despite no organic abnormality of digestive tract function. Due to these disabilities, the resulting malnutrition, and the slow recovery, it is often difficult for elderly individuals to reintegrate into society. Secondary or incorrect nutritional management increases complications, decreases physical function and worsens the prognosis. Previous statistical research suggests that in-hospital mortality is significantly higher among hospitalised patients aged ≥65 years who ingest less than half of their caloric needs. Therefore, appropriate nutritional management from an early stage is essential for elderly individuals. Moreover, functional excretion disorders, dementia, and sarcopenia (muscle-wasting disease) are attracting attention as pathological conditions unique to elderly individuals, and it is essential to undergo rehabilitation early with nutritional management. Being elderly does not preclude nutritional management, and it is necessary to reconsider appropriate nutritional therapy even in the terminal stage and in advanced physical and mental illnesses. This review explores the relationship between dietary intake and FGIDs, with a focus on elderly adults.

## 1. Introduction

### 1.1. Definition of Functional Gastrointestinal Disease

Functional gastrointestinal disease (FGID) is characterised by chronic or recurrent gastrointestinal symptoms, the absence of organic lesions on laboratory examination, and symptomatic dysfunction [1]. FGIDs include irritable bowel syndrome (IBS), functional bloating, constipation, diarrhoea, and unspecified available bowel disease [1,2,3]. Symptoms of IBS, the classic FGID, include abdominal pain, discomfort, and associated bowel abnormalities [3]. The mechanism of development of irritable bowel syndrome is illustrated in Figure 1. Physical and mental stress play a major role in the onset of IBS and the worsening of symptoms. The disease is etiologically based on an individual’s innate personality or the environment in which the individual grew up, which could cause the intestines to become sensitive. When physical and mental stress is added to this situation, intestinal dysfunction occurs. The intestines become spasmatic and contract excessively or become unable to relax, causing abnormal movement. Additionally, the brain and intestines become sensitive, causing sensory abnormalities. Symptoms of hypersensitivity syndrome are thought to arise from abnormalities in movement and sensation [1,2,3]. Rome IV diagnostic criteria are used to diagnose IBS [4]. According to the Rome IV diagnostic criteria, IBS is characterised by bowel or bowel-frequency-related abdominal pain occurring at least one day per week for the past three months. In addition to the persistence of symptoms for six months, two or more of the above three items associated with changes in stool shape can be used to diagnose IBS patients [4].

Irritable bowel syndrome (IBS) is characterised by abdominal symptoms such as diarrhoea, constipation, other bowel movements, abdominal pain and bloating, even though there are no organic abnormalities such as ulcers or tumours in the large or small intestine (Figure 1). Physical and mental stress are greatly involved in the onset of IBS or the worsening of symptoms. The disease is etiologically based on an individual’s innate personality or the environment in which the individual grew up, which could cause the intestines to become sensitive. With the addition of physical and mental stress, intestinal dysfunction occurs. The intestines may spasm, contract excessively or be unable to relax, resulting in movement abnormalities. Additionally, the sensations in the brain and intestines become sensitised, and sensory abnormalities occur. Hypersensitivity syndrome is thought to result from motor and sensory abnormalities.

Subtypes of IBS can be divided into IBS with diarrhoea (IBS-D), IBS with constipation (IBS-C), mixed IBS (IBS-M), and unclassifiable IBS (IBS-U). These subtypes are thought to be helpful in clinical practice and therapy. The Rome IV criteria assess only subjective symptoms [4]. Therefore, the diagnostic criteria for Rome IV are challenging to apply to patients with unconsciousness or cognitive impairment [5]. Therefore, professional medical examination and clinical judgement may be needed. Furthermore, the stool morphology in patients with IBS varies from watery to hard [6], suggesting that the gastrointestinal transit time is associated with stool morphology [7,8,9]. However, there are also reports that IBS patients’ frequency and gastrointestinal transit time are similar to those of healthy individuals [10,11]. Gastrointestinal transit times may require further scientific validation.

IBS has a negative impact on daily life, so it is essential to improve lifestyle habits to alleviate symptoms [12,13]. As described above, IBS causes intestinal motility disorders such as constipation, diarrhoea, and abdominal pain associated with abnormal defecation and defecation symptoms. This differs for each IBS subtype. According to the Bristol stool form scale (BSFS), the four subtypes of IBS are IBS-C, IBS-D, IBS-M, and IBS-U. Nevertheless, depending on the subtype, IBS is also a risk factor for faecal incontinence and has been reported to reduce patients’ quality of life (QOL) [14,15,16].

### 1.2. Association between Food Intake and Gastrointestinal Motility Abnormalities

Patients with excretion disorders should be instructed to refrain from ingesting caffeine, citrus fruits, spicy foods, and alcohol as dietary and lifestyle guidance [17]. Dietary fibres such as psyllium have been reported to reduce faecal incontinence by improving stool quality [18]. In addition to antidiarrhoeal drugs such as loperamide hydrochloride, dietary fibre intake has also been reported to improve faecal incontinence [19].

On the other hand, a study was conducted in which elderly stroke patients who had weakened control of bowel movements were instructed to change their diet and fluid intake. Afterwards, it was found that the number of regular bowel movements increased; their faecal incontinence remained significant and did not improve [20]. The teaching of bowel habits is an essential factor in preventing faecal incontinence. If the rectal sensation is normal, it is recommended that individuals go to the toilet as soon as they feel the urge to defecate. On the other hand, if the rectal sensation is reduced, a planned defecation attempt without the desire to defecate can significantly improve faecal incontinence [21,22].

In elderly individuals, the loss of rectal sensation may cause stool to accumulate in the rectum, and leaky incontinence may occur if stool continues to accumulate in the rectum without the urge to defecate. For such patients, defecation habit training (stress defecation), in which they go to the toilet and defecate twice a day (approximately 30 min after breakfast or dinner), even if they do not feel the urge to defecate, may be effective [21,22]. Educational guidance and advice on defecation from nurses reduce faecal incontinence in older adults and are also beneficial for caregivers [22]. Inadequate management of voiding disorders can lead to dermatitis on the buttocks, including erythema, erosions, and ulcers. A moisturising and protective skin care regimen using mildly acidic cleansers and skin dressings reduces the incidence of faecal-incontinence-related dermatitis [22].

To summarise the above discussion, nutritional therapy, defecation habits and skilled care are essential to prevent dysuria, and it is considered necessary to actively implement these measures.

### 1.3. Food Intake Challenges in the Elderly

The risk of malnutrition in elderly patients is high, and a nutritional assessment should always be carried out with malnutrition in mind. In general, elderly individuals have a lower basal metabolism, decreased food intake, and decreased physical activity. Physiological deterioration of taste and smell, social factors such as living alone, and psychophysiological factors such as dementia and depression also affect them. In addition, diseases such as malignant tumours, infectious diseases, heart failure, and respiratory failure greatly hinder dietary intake. The medications used are often the cause of anorexia. Many of these risks contribute to the high incidence of malnutrition in elderly individuals.

Regarding the incidence of malnutrition, a previous study [23] found that it affects 5% of outpatients, 20% of inpatients, and 37% of institutionalised patients. Since malnutrition in elderly individuals is often overlooked, multiple nutritional indicators are used for evaluation. Nutritional assessment should not only assess the current dietary status but also anticipate future risks. Malnutrition is easily ignored because elderly individuals often have comorbidities, dehydration, oedema, and other factors that affect multiple nutritional endpoints. In addition to subjective factors, it is desirable to use various blood indicators, such as serum albumin levels, mental characteristics, and physical activity. Recently, the Geriatric Nutritional Risk Index (GNRI) [24] and Mini-Nutritional Assessment (MNA^®^) [25] have been proposed as nutritional assessment tools for elderly individuals.

### 1.4. Indications for Nutritional Therapy in the Elderly Population

Indications for nutritional therapy are fasting for three days or more, insufficient oral intake for seven days or more, progressive weight loss (5% or more in 1 month, 10% or more in 6 months), a BMI less than 18.5, and serum albumin levels below 3.0 g/dL [26,27]. In particular, elderly individuals are inherently poor in reserve capacity and have fragile physical functions. Recovery from illness is slow, and the reintegration capacity is limited. Therefore, it is necessary to detect nutritional risks at an early stage and start preventive nutritional therapy before malnutrition develops. Since there is no study that provides a sufficient basis for determining this criterion, future research is expected.

### 1.5. Proper Nutritional Dosage

Additionally, many elderly individuals are prone to micronutrient deficiencies and require aggressive supplementation [28,29]. Phosphorus is essential in patients with malnutrition, and previous studies found hypophosphatemia in 14.1% of elderly hospitalised patients. In the group with hypophosphatemia, the length of hospitalisation was significantly longer, and the mortality rate was considerably higher; three-fold, suggesting the involvement of refeeding syndrome [30]. Administration of vitamin B (B1: thiamine) is necessary to prevent Wernicke encephalopathy associated with carbohydrate administration.

This review aims to explore the relationship between dietary intake and FGIDs, with a focus on elderly adults.

## 2. Materials and Methods

In this study, we provide an overview of nutritional therapy implementation and evaluation methods for diseases that elderly individuals are prone to develop. From the literature thus far, the authors discuss existing knowledge, challenges, and prospects. Cited literature was searched in PubMed and Centralblatt für die gesammte Medicine (https://www.jamas.or.jp), and references were obtained. Most of the papers were published in English, but a few references were also made to Japanese literature. From these documents, we extracted articles with scientific knowledge that have already been published in academic journals. The search keywords for removing these papers were “elderly”, “nutritional therapy”, gastrointestinal motility, “functional digestive disease FGID”, and “daily lifestyle habits”. This is a review of the literature on these subjects. There were no restrictions on the date of publication, sample size, study design, or the age of subjects, and only published articles that reported scientific knowledge and consensus were cited.

## 3. Results and Discussion

### 3.1. Lifestyle Improvements and Dietary Guidance Are Effective for Chronic Constipation

There are many different types of defecation disorders in elderly individuals. It has been suggested that chronic constipation can be improved by lifestyle changes and dietary guidance. It has been reported that there is not necessarily a correlation between chronic constipation and dietary fibre intake and that dietary fibre intake is effective only when the intake is insufficient [31]. Limiting the intake of fermentable, poorly absorbed short-chain carbohydrates such as oligosaccharides, disaccharides, monosaccharides, and polyols has been shown to help reduce chronic constipation and lower the risk of constipation-predominant IBS. However, it has also been reported that the intake of short-chain carbohydrates leads to increased gas production due to fermentation and has only a small effect on chronic constipation [32]. However, among fermentable dietary fibres, partially hydrolysed guar gum (PHGG), which is often reported to be used in Japan, significantly increases the frequency of defecation in cases of chronic constipation [33,34,35,36] and has been shown to significantly reduce the laxative amounts used [37]. Butyric acid, a short-chain fatty acid produced by fermentation of dietary fibre, has been reported to promote the production of serotonin, which increases intestinal motility [38]. For chronic constipation, an RCT conducted with patients consuming kiwifruit, prunes, and psyllium (plantain) found that all the foods increased the natural defecation rate and frequency of defecation to the same degree. Psyllium has also been shown to be effective in treating chronic constipation [39]. Furthermore, in an RCT of Japanese people with chronic constipation who ingested prunes and a placebo, the prune intake group showed an increase in the proportion of normalised stool shapes and symptom score related to the number of hard stools and defecation urges on the Gastrointestinal Symptom Rating Scale (GSRS) [40]. On the other hand, in a study examining the effects of kiwifruit intake on intestinal function in healthy subjects, MRI revealed an increase in the volume of the small intestine and ascending colon, a significantly increased frequency of defecation, and a decreased number of defecations on the Bristol Stool Form Scale (BSFS), suggesting that kiwifruit intake may increase intestinal water [41].

Regarding the effectiveness of exercise therapy for chronic constipation, a meta-analysis reported that aerobic exercise is particularly effective in improving symptoms [42]. However, since the target literature contains many biases, we believe that a more-rigorous examination is necessary to evaluate the effect of exercise therapy on chronic constipation. A large-scale data analysis has been conducted on dietary fibre intake and its effect on improving constipation according to physical activity and gender [43]. For people with high physical activity, there was a relationship between dietary fibre intake and stool consistency, but for people with low physical activity, there was no relationship between dietary fibre intake and stool consistency. This suggests that dietary fibre intake is less effective for chronic constipation in people with low physical activity. In an RCT in which young women were divided into groups with and without trunk stretching, abdominal muscle strength increased in the stretching group, but no change was observed in stool transit time to the large intestine [44]. In contrast, a different RCT reported that abdominal wall massage for 15 min a day, 5 times a week, was effective in improving chronic constipation [45]. To test the effect of water intake on chronic constipation, another RCT divided patients into two groups: those who consumed sufficient dietary fibre (25 g) and a large amount of water (2.1 L on average) and those who had a normal amount of water intake (1.1 L on average); it was shown that the frequency of defecation was significantly increased in the group that drank copious amounts of water [46]. On the other hand, four reports comparing water intake between a chronic constipation group and a control group found no difference between the two groups [47], and the effectiveness of water intake for chronic constipation has not been demonstrated [48].

### 3.2. Lifestyle Improvements, Dietary Guidelines, and Bowel Habit Guidance Are Effective for Faecal Incontinence

As dietary and lifestyle guidance for faecal incontinence, it is important to instruct patients to refrain from consuming caffeine, citrus fruits, spicy foods, and alcohol, which have the effect of softening stool [17]. Dietary fibre supplements such as psyllium have been reported to reduce faecal incontinence by improving stool quality [18]. Additionally, an RCT showed that faecal incontinence can be improved by taking dietary fibre in addition to taking antidiarrhoeal drugs such as loperamide hydrochloride [19]. On the other hand, an RCT in which elderly stroke patients with decreased physical strength were instructed to change their diet and fluid intake to regulate their bowel movements found that although the number of normal bowel movements increased, faecal incontinence was not significantly reduced, and the patients reported no improvement [20].

Guidance on bowel habits is an important element in the treatment of faecal incontinence. If rectal sensation is normal, it is recommended that individuals go to the toilet as soon as possible without holding back when they feel the urge to defecate [21,22]. On the other hand, if rectal sensation is decreased, faecal incontinence can be significantly improved by systematically attempting to defecate even if there is no desire to defecate [21,22]. That is, among individuals with spinal cord disorders and elderly people, those with decreased rectal sensation may experience overflow faecal incontinence. It is thought to be effective to train these patients in improving their bowel habits. This can be accomplished by their going to the toilet and defecating twice a day, 30 min after breakfast and dinner, even if they have no desire to defecate [22].

### 3.3. How Do Manage Nutrition for Elderly Individuals?

Oral intake is the first choice, and even for elderly individuals who try to increase food intake and promote supplementary meals, oral and enteral nutrition must be prioritised if there are no problems with the digestive tract. The reason for this is that if gastrointestinal function remains, oral intake or enteral nutrition should be performed as much as possible to prevent atrophy of the gastrointestinal tract. Preventing a decline in gastrointestinal function also helps prevent malnutrition and infections. Prior research on oral nutritional supplements (ONSs), which are supplements for enteral nutrition, has shown that ONSs reduce the mortality rate and complication rate in malnourished patients and has been found to be useful [49]. It is first desirable to promote oral intake, including a review of the diet itself. Although an ONS is taken in small amounts, it is high in calories and contains most of the necessary nutrients; consequently, it is the basis for maintaining health when people cannot eat or when the amount of food they eat is reduced, until they can receive enough nutrition from meals. An ONS supports the body as a nutritional energy source. However, compliance with ONSs is by no means good, and it is necessary to devise measures such as providing a variety of flavours so that patients do not become bored, and efforts such as always having staff available to talk to them are necessary. If the oral intake, including ONS administration, is inadequate, tube feeding is suggested. Furthermore, when introducing artificial nutrition, it is essential to consider whether it will benefit the patient, contribute to the outcome and recovery, and contribute to the maintenance and improvement of QOL, as well as whether the benefits outweigh the risks and what lifestyle is desired by the individual. It is necessary to consider whether the patient complies with the standard, whether there are medical resources that can appropriately manage it, and whether changes in the medical environment will be disadvantageous to the patient when long-term tube feeding is adopted. Since IBS is not an organic disease, improving eating habits and reducing stress can improve symptoms. Because IBS is a disease that lasts for a long time, it is very important for patients to try to improve their lives by adjusting their regular eating habits and lifestyle rhythms to eliminate stress.

Second, tube feeding should be considered if oral intake is inadequate. The effects of tube feeding on elderly patients have varied among reports. A previous study [50] prospectively observed the use of percutaneous endoscopic gastrostomy (PEG) in elderly subjects and found that serum albumin and transthyretin levels were significantly elevated, but the QOL scores were unchanged. Another study [51] prospectively observed elderly patients who underwent PEG and reported that at least 70% showed no noticeable improvement in functional or nutritional aspects. In addition, in a study of 52 cases in which tube feeding was introduced at the end of dementia, significant weight gain was obtained in the tube-feeding group compared with a control group without tube feeding [52]. Aspiration pneumonia and physical restraint tended to be more common, and that problem has been pointed out. Another study retrospectively examined PEG patients with dementia and reported that none of the elderly patients experienced functional improvement after PEG, and no significant nutritional improvement was observed [53]. In addition, many elderly individuals have cognitive and mental problems and are prone to self-removal of nasal catheters. A study of seven elderly patients reported that 60% of patients voluntarily removed the nasal catheter, and 40% of patients developed aspiration pneumonia in the first two weeks of tube feeding [54]. Many cases of indwelling nasal catheters at medical sites include physical restraint, mainly of the upper extremities, to prevent self-removal of the catheters. For long-term tube feeding, a transition to gastrostomy should be considered to maintain and improve the QOL.

Third, if dysphagia is observed, tube feeding should be introduced as early as possible. On the other hand, when neurologic dysphagia is present, its benefits and needs are clear, and tube feeding is a good indication [55,56,57,58,59]. Inappropriate fasting can lead to poor nutritional status, and oral aspiration can lead to pneumonia. This leads to more extended hospital stays and increased mortality, so it is essential to introduce tube feeding as early as possible [60,61,62]. Some studies have reported that enteral feeding using a gastrostomy yields better clinical outcomes than enteral feeding using a nasogastric catheter in the acute phase of cerebral infarction with dysphagia [63,64,65]. It is safer to choose a nasal catheter during the acute stage and gastrostomy after entering the stable phase. In addition, dysphagia due to cerebral infarction has been reported to be temporary, and 4–29% of patients are able to return to oral intake after 4–31 months [63,66,67,68,69,70]. It is essential to continue swallowing training and regular assessment of swallowing function even after tube feeding is introduced.

Finally, parenteral nutrition is suggested when the gastrointestinal tract is difficult to access. Regarding parenteral nutrition, total parenteral nutrition (TPN), in particular, has the risk of various complications and is a costly and invasive procedure that should be reserved for intolerant patients. Recently, an increasing number of peripherally inserted central catheters (PICCs) have been shown to contribute to the reduction of complications such as catheter-related bloodstream infections (CRBSIs), and thrombosis [71] is expected to spread further. Additionally, a randomised controlled trial (RCT) examining the effects of PPN in patients with inadequate enteral nutrition (EN) after acute illness found that transthyretin and CD4 levels and the physical function improved and that peripheral parenteral nutrition (PPN) was safe and effective [72]. However, it has been reported that although patient selection for PPNs was generally appropriate, there were many problems with the content of the prescription [73].

### 3.4. Nutritional Therapy for Elderly Individuals with Functional Gastrointestinal Disease

Eating meals is the best nutritional management method, but the percentage of hospital diets that are adequately ingested is by no means high [74]. Decreased food intake due to the primary disease and problems with the hospital food, such as the taste, menu, and serving time, are important factors [75]. Therefore, when it is impossible to obtain the necessary amount of nutrients through oral intake alone, nutritional therapy using parenteral or enteral nutrition is needed. Nutritional therapy should be considered if the energy expenditure or the ability to consume less than 60% of the necessary amount are expected to continue for more than a week. The important question is what kind of pathology and judgement criteria should be used to select between parenteral and enteral nutrition. The basic principle is “if the gut works, use the gut”. The reason is that enteral nutrition is more physiological than parenteral nutrition and maintains the original functions of the gastrointestinal tract, such as digestion and absorption, and the functions of the intestinal immune system. Intestinal mucosal atrophy occurs when the gastrointestinal tract is not used during parenteral nutrition, which is a factor in bacterial translocation (BT) [76,77], whereas enteral nutrition has been confirmed to reduce the intestinal mucosal integrity (homeostasis) [78]. Thus, BT associated with parenteral nutrition under fasting conditions has been approved by many animal experiments using burn models.

On the other hand, in humans, although intestinal mucosal atrophy is observed during parenteral nutrition under fasting conditions [79], it has been reported that the change is minor [80]. It is unclear whether this could be a factor [81,82]. There is also a view that infectious complications do not increase even in TPN cases if an appropriate amount of energy is administered [83]. However, many studies have demonstrated intestinal mucosa atrophy when the intestinal tract is not used (e.g., fasting), which leads to a decrease in mechanical barrier function and immunological barrier function [84,85].

In clinical comparisons between parenteral nutrition and enteral nutrition, it is also true that the incidence of infectious complications is lower with enteral nutrition than with parenteral nutrition [86,87]. This is because the integrity of the intestinal mucosa is maintained, and the mechanical and immunological barrier functions are maintained by administering nutrients into the gastrointestinal tract. Early enteral nutrition is recommended for these benefits, especially in patients with burns and severe acute pancreatitis [88,89,90,91,92]. Enteral nutrition is theoretically contraindicated, and parenteral nutrition is limited for cases of general peritonitis, intestinal obstruction, refractory vomiting, paralytic ileus, refractory diarrhoea, and active gastrointestinal bleeding.

In recent years, the usefulness of enteral nutrition has been highly evaluated, and it is indicated for many diseases. However, depending on the target disease, the superiority of enteral nutrition over parenteral nutrition is not necessarily proven, and there are studies showing that enteral nutrition and parenteral nutrition are equally helpful [93,94]. Therefore, it is essential to understand the characteristics of parenteral nutrition, enteral nutrition, and both nutrition methods, and to select the appropriate intervention according to the disease state.

### 3.5. Nutritional Therapy for Elderly Individuals with Reduced Cognitive Motor Function

Dementia is a specific disease in which cognitive motor function is reduced. Dementia causes anorexia (sometimes drug-induced), poor food intake, depression, and often malnutrition. When dementia becomes severe, dysphagia is also complicated. It was reported that ONS was well tolerated and beneficial for malnutrition due to dementia, resulting in weight gain [95,96]. On the other hand, the effect of tube feeding on improving nutritional parameters was mixed [97], and there were many reports that it did not contribute to the improvement of physical function or the survival rate. Regardless of whether the catheter is a nasogastric catheter or a gastrostomy, there are obvious problems associated with catheter placement, including discomfort. The pros and cons of nutritional therapy must be carefully weighed on a case-by-case basis, considering ethical and social aspects. The points to be considered are (1) the patient’s anticipated willingness to receive artificial nutrition; (2) the severity of the disease itself; (3) the patient’s prognosis; (4) the presence or absence of tube feeding; (5) QOL, complications and physical limitations due to tube feeding; and (6) the degree of physical activity of the patient. For severe dementia patients who are bedridden or who have difficulty with communication due to the need for total assistance, nutritional therapy is limited. However, it is necessary to confirm the ethical and social circumstances of administering PEG to elderly individuals with dementia and to decide individually whether to apply nutritional therapy considering the family’s wishes.

### 3.6. Nutritional Therapy for the Elderly Should Be Combined with Early Rehabilitation

Elderly individuals are prone to sarcopenia, which is characterised by decreased muscle mass and strength due to reduced nutrient intake and decreased physical activity. Nutrition-related factors include inadequate energy and protein intake, and the incidence of sarcopenia increases with age. Prevention of sarcopenia is essential. Once sarcopenia occurs, it takes time to recover. When an elderly person is diagnosed with malnutrition, nutritional therapy should be started as early as possible, and exercise therapy should also be used as a countermeasure against sarcopenia. Merely performing nutritional therapy will not increase muscle mass. Although there is no age-related difference in the rate of protein synthesis from administered amino acids, exercise therapy is necessary to increase muscle mass [98]. Exercise therapy combined with nutritional therapy is helpful as a treatment for sarcopenia in the elderly [99], and protein supplementation during muscle training increases muscle mass and strength, according to a meta-analysis [100]. These studies suggest that it should be noted that if rehabilitation is performed without nutritional therapy, sarcopenia will be aggravated [98,99,100].

## 4. Conclusions

Japan is ahead of the rest of the world in facing a superaged society, and nutritional therapy management for the elderly population is essential in Japan. Elderly individuals often have multiple diseases and are prone to malnutrition. In addition, functional constipation, diarrhoea, faecal incontinence, etc., may occur even though there is no organic abnormality in the function of the digestive tract. These disorders, the malnutrition resulting from them, and slow recovery from disease and malnutrition often make it difficult to reintegrate into society. Secondary or incorrect nutritional management increases complications, decreases physical function and worsens the prognosis. In-hospital mortality was reported to be significantly higher in hospitalised patients aged 65 years or older who ate less than half of their calorie requirements than in those who did not. Therefore, appropriate nutritional management from an early stage is essential for elderly individuals. In recent years, the concepts of functional excretion disorders, dementia, and sarcopenia have attracted attention as pathological conditions unique to elderly individuals, and it is essential to perform rehabilitation from an early stage together with nutritional management. When functional excretory disorders in elderly people have subjective symptoms, it is necessary to understand the excretory status according to the individual. It is necessary to consider the method, form, and timing of nutritional administration depending on the excretory disorder. Dementia and sarcopenia often have no subjective symptoms. Therefore, collecting more rigorous and objective information for nutritional management is necessary. On the other hand, bedridden elderly individuals and patients with severe dementia are increasingly being fed by tube feeding, and the problem of excretion disorders and a decrease in social activities has become a social problem peculiar to Japan. Being elderly does not preclude the application of nutritional management, and it is necessary to reconsider appropriate tube-feeding nutrition therapy for patients with dementia in the terminal stage and in those with advanced physical and mental illnesses.

Limitations of this study include no restrictions on publication date, sample size, study design, or subject age, and only published articles reporting scientific knowledge and consensus were cited. Also, although it refers to nutritional management for older adults in Japan, many of the mentioned documents are from foreign countries, so it will be necessary to carefully scrutinize the literature before applying it.

## Figures and Tables

**Figure 1 healthcare-11-02829-f001:**
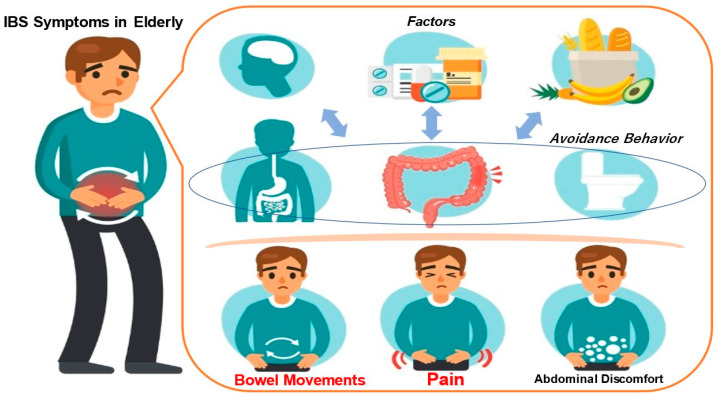
Mechanism of IBS manifestation in elderly individuals.

## Data Availability

As per the conditions outlined in the identifiable information included in the data file and survey materials, these items have not been made available.

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
