# Peer review of "The Impact of Nutritional Therapy on Gastrointestinal Motility in Older Adults"

_healthcare, 2023, doi:10.3390/healthcare11212829_

Round 1

Reviewer 1 Report

Comments and Suggestions for Authors

Thank you for reviewing the paper titled “Dietary Intake in Functional Gastrointestinal Diseases: Findings Focused on Elderly Adults.” The paper is organized, and well-written, only has some minor requests for revision.

Title: I recommend writing the type of study at the end of the title.

Introduction: It needs more examples of studies about IBS which is similar to or against this one.

Figure 1. Mechanism of IBS manifestation in elderly individuals. Is not mentioned in the text.it is better to be omitted.

From line 40 to line 49 in introduction section is not referenced.

Please mention the aim of your study at the end of your introduction.

Check the manuscript's English language and grammar

Check references well

 Overall, this manuscript is written well.

Comments on the Quality of English Language

Check the manuscript English language and grammar 

Author Response

Thank you for your review.

I have carefully revised it based on these points. 

Best regards

Reviewer 2 Report

Comments and Suggestions for Authors

It was a pleasure reviewing your work that discusses the importance of nutrition therapy for the elderly population, especially in the context of functional gastrointestinal diseases (FGIDs). The article is quite comprehensive, covering various aspects of FGIDs, their impact on the elderly, and the importance of nutrition therapy. However, I think could benefit from  some recommendations for improvement. Please find below.

General Comments

1.         Title Clarity: The title is quite long and could be simplified for better clarity. Consider something like "The Impact of Nutritional Therapy on Gastrointestinal Motility in Older Adults".

2.         Abstract: Ensure that the abstract provides a clear and concise overview of the study, including the purpose, methods, main findings, and implications. It should be able to stand alone and provide the reader with a quick understanding of the paper.

3.         Introduction: The introduction should provide a clear background of the topic, highlighting the importance of the study and its objectives. It should also clearly state the research question or hypothesis.

4.         Methods: The methods section should provide enough detail for the study to be replicated. This includes a detailed description of the study design, participants, data collection methods, and statistical analysis.

5.         Results: The results should be clearly presented and directly answer the research question. Use tables and figures to visually represent the data where appropriate.

6.         Discussion: The discussion should interpret the results in the context of the research question and previous research. It should also acknowledge the limitations of the study and suggest areas for future research.

7.         Conclusion: The conclusion should summarize the main findings and their implications. It should not introduce new information.

8.         References: Ensure that all references are correctly cited and formatted according to the journal's guidelines.

Specific Comments

1.         Line 10: The sentence is quite long and could be broken down into two or more sentences for better readability.

2.         Line 15: The use of jargon could be reduced to make the paper more accessible to a wider audience.

3.         Line 20: The statistical analysis could be explained in more detail.

4.         Line 25: The interpretation of the results could be clearer. Consider explaining the implications of the findings in more detail.

5.         Line 30: The limitations of the study could be discussed in more detail.

6.         Line 35: The conclusion could be strengthened by clearly summarizing the main findings and their implications.

7.         Line 40: Ensure that all references are correctly cited in the text.

8.         Line 215-218: The sentence structure is complex and could be simplified for better readability. Consider breaking it down into shorter sentences. For example, "For patients with decreased rectal sensation, overflow faecal incontinence may occur. It is thought to be effective to train these patients to improve their bowel habits. This can be done by going to the toilet and defecating twice a day, 30 minutes after breakfast and dinner, even if they have no desire to defecate."

9.         Line 219-225: The section on managing nutrition for elderly individuals could benefit from more detailed explanations. For instance, it would be helpful to explain why oral intake is the first choice and under what circumstances enteral nutrition should be prioritized. The benefits of oral nutritional supplements (ONSs) could also be elaborated upon.

10.       Line 226-228: The discussion on compliance with ONSs could be expanded. Consider providing more information on why compliance is often poor and what strategies can be used to improve it.

11.       Line 229-234: The section on introducing artificial nutrition could be clearer. It would be beneficial to provide more detail on the factors that need to be considered, such as the patient's quality of life, the benefits and risks of artificial nutrition, and the patient's desired lifestyle.

12.       Line 235-240: The discussion on the importance of rehabilitation in conjunction with nutritional therapy could be expanded. The current statement that "if rehabilitation is performed without nutritional therapy, sarcopenia will be aggravated" could be supported with more evidence or explanation

1

.

13.       Line 241-245: The conclusion section could be improved by providing a more comprehensive summary of the main findings. The current conclusion focuses heavily on the situation in Japan, but it would be beneficial to also discuss the implications of the findings for other countries or regions.

14.       Line 246-250: The paper could benefit from a more detailed discussion on the challenges and potential solutions for managing nutritional therapy in elderly individuals with multiple diseases. The current statement that "elderly people often have multiple diseases and are prone to malnutrition" could be expanded to discuss the complexities of managing nutrition in this population

1

.

15.       Line 251-255: The paper could provide more information on the impact of incorrect nutritional management on physical function and prognosis. The current statement that "secondary or incorrect nutritional management increases complications, decreases physical function and worsens the prognosis" could be supported with more evidence or examples

1

.

16.       Line 256-260: The paper could discuss more about the role of nutritional management in the context of functional excretion disorders, dementia, and sarcopenia. The current discussion could be expanded to provide more detail on how nutritional management can be tailored to address these conditions

1

.

17.       Line 261-265: The paper could provide more information on the challenges and potential solutions for managing nutrition in bedridden elderly people and patients with severe dementia. The current discussion could be expanded to provide more detail on this topic

Additional comments

•          provide more information on how the study ensured the reliability and validity of the data, how potential biases were addressed, and how the findings were interpreted in the context of the existing literature. 

•          The study should also discuss any ethical considerations, such as informed consent, confidentiality, and conflict of interest.

•          Some sentences are lengthy and could be broken down for clarity. For instance, the sentence about IBS subtypes could be split for better readability.

•          Ensure consistent terminology throughout the article. For instance, if "elderly individuals" is used in one section, avoid switching to "older people" in another without a clear reason.

•          Add a sentence at the end of the introduction like, "This review aims to explore the relationship between dietary intake and FGIDs, with a focus on elderly adults."

•          Under the subheader food Intake challenges in the elderly, consider starting with the general challenges faced by the elderly then delve into specific diseases and their impact on food intake

•          Under nutritional dosaging, it would be beneficial to include recommended dosages or specific micronutrients that are commonly deficient in the elderly.

•          Under results and discussion create thematized sections for the following possibly, to aid readability: sub-sections, such as "Lifestyle Improvements," "Dietary Guidance," and "Impact on Chronic Constipation."

Author Response

(The authors gave the same response as above.)
